# Testing the Domino Theory of Gene Loss in *Buchnera aphidicola*: The Relevance of Epistatic Interactions

**DOI:** 10.3390/life8020017

**Published:** 2018-05-29

**Authors:** David J. Martínez-Cano, Gil Bor, Andrés Moya, Luis Delaye

**Affiliations:** 1Departamento de Ingeniería Genética, CINVESTAV Irapuato, Km. 9.6 Libramiento Norte Carretera Irapuato-León, 36821 Irapuato, Guanajuato, Mexico; boost07@yahoo.com; 2CIMAT, A.P. 402, Guanajuato 36000, Gto., Mexico; gil@cimat.mx; 3Fundación para el Fomento de la Investigación Sanitaria y Biomédica de la Comunitat Valenciana (FISABIO)-Salud Pública, Avenida de Catalunya 21, 46020 València, Spain; andres.moya@uv.es; 4Institute for Integrative Systems Biology, Universitat de València, Calle Catedrático José Beltrán 2, 46980 Paterna, València, Spain

**Keywords:** endosymbiotic bacteria, genome reduction, gene interactions, correlated evolution

## Abstract

The domino theory of gene loss states that when some particular gene loses its function and cripples a cellular function, selection will relax in all functionally related genes, which may allow for the non-functionalization and loss of these genes. Here we study the role of epistasis in determining the pattern of gene losses in a set of genes participating in cell envelope biogenesis in the endosymbiotic bacteria *Buchnera aphidicola*. We provide statistical evidence indicating pairs of genes in *B*. *aphidicola* showing correlated gene loss tend to have orthologs in *Escherichia coli* known to have alleviating epistasis. In contrast, pairs of genes in *B*. *aphidicola* not showing correlated gene loss tend to have orthologs in *E*. *coli* known to have aggravating epistasis. These results suggest that during the process of genome reduction in *B*. *aphidicola* by gene loss, positive or alleviating epistasis facilitates correlated gene losses while negative or aggravating epistasis impairs correlated gene losses. We interpret this as evidence that the reduced proteome of *B*. *aphidicola* contains less pathway redundancy and more compensatory interactions, mimicking the situation of *E*. *coli* when grown under environmental constrains.

## 1. Introduction

Mutualistic endosymbiotic bacteria from insects show the most reduced genomes of all cells [1]. The genomes of these bacteria are the outcome of an evolutionary process driven by the symbiotic interaction with its host [2]. One of the best studied cases of genome reduction among these prokaryotes is that of *Buchnera aphidicola*, the endosymbiotic bacteria from aphids [3]. The basis of this mutualistic interaction is nutritional. *B*. *aphidicola* provides essential amino acids lacking in the diet of its host [4,5,6,7,8,9].

Genome reduction in these bacteria is a complex process that is only partially understood. The sequence of events leading to small genomes has been described by sequencing the genomes of bacteria in different stages of genome reduction [10]. However, the role played by evolutionary forces like selection and genetic drift during genome reduction have been more difficult to untangle. It is generally accepted that these genomes accumulate slightly deleterious mutations through Muller’s ratchet process [11,12,13]. However, there is clear evidence that selection also plays a role in the evolution of these cells [14,15,16,17].

One hypothesis proposed to explain how these genomes shrink is the *domino theory* of gene loss [18]. The domino theory is based on the premise that the products of genes interact and that this interaction affects the fitness of the organism. Accordingly, the domino theory states that when some particular gene loses its function and severely cripples or causes the total loss of a cellular function, selection will relax in all functionally related genes, which may trigger the non-functionalization and loss of these genes.

There is evidence showing that genes whose products interact, tend to be lost together in symbiotic bacteria. In *Paulinella chromatophora*, genes coding for the ABC phosphate transporter pstSACB provide an example of correlated loss [19]. The ABC phosphate transporter is coded by four different genes: *pstS*, *pstC*, *pstA*, and *pstB*. The *pstS* gene codes for a phosphate binding periplasmic protein; *pstB* is an ATP binding protein located on the cytoplasmic side of the membrane; while *pstC* and *pstA* are two cytoplasmic transmembrane permeases [20]. The chromatophore of *P. chromatophora* M0880/a contains all four genes, with *pstC*, *pstA*, and *pstB* located in one locus and *pstS* found in a very different location in the genome. However, in the chromatophore of another strain (*P. chromatophora* FK01), precisely these four genes were jointly lost. Another example of coordinated gene loss is provided by *Mycobacterium leprae*. For instance, it was suggested that the loss of genes coding for the two-component system involved in hypoxia response (DevR and DevS) triggered a domino effect of gene loss in this bacterium [21]. Lastly, in *B*. *aphidicola* BCt it was found that the gene *pgi* coding for glugose-6-phosphate isomerase has been lost. As a consequence, this endosymbiont can only use fructose-6-phosphate for glycolysis. Associated to this loss, *B*. *aphidicola* BCt lacks *pgl* and *zwf* genes of the oxidative pentose pathway but retains the genes necessary to obtain ribose-5-phosphate from fructose-6-phosphate [22].

Gene loss has been carefully studied in a set of five different *B*. *aphidicola* genomes [22]. Thanks to extreme synteny (gene order has been conserved for at least 50 million years) and almost total lack of horizontal gene transfer, it is relatively easy to identify orthologs and reconstruct the history of gene losses among the *Buchnera* [6,23]. In accordance with the domino theory, it was found that more recent gene losses tend to occur in pathways already affected by previous gene losses [22]. 

An intriguing question is the relationship between the domino theory and epistasis. One definition of epistasis (*functional epistasis*) addresses “the molecular interactions that proteins have to one another” [24]. This definition of epistasis is expressed in the same terms as the domino theory of gene loss. Accordingly, epistasis can be measured in terms of the observed versus the expected fitness effects of mutations where these mutations are independent, for instance in two different loci. The expected outcome is calculated by using a multiplicative model whereby ω*_i_* and ω*_j_* are the fitness effects of deleting loci *i* and *j* independently; and ω*_ij_* is the fitness effect of deleting both loci at the same time. Then, epistasis is defined as: ε*_ij_* = ω*_ij_* − ω*_i_*ω*_j_*. The sign of ε*_ij_* determines if there is alleviating (positive) or aggravating (negative) epistasis. Positive epistasis is also known as alleviating gene interaction, while negative epistasis is known as aggravating gene interaction.

Babu et al. (2011) [25] studied epistatic interactions among genes involved in cell envelope biogenesis pathways in *Escherichia coli*. They studied the fitness effects of all possible digenic mutant combinations of 821 genes. These genes were divided into 128 essential biosynthetic genes (represented by partial loss of gene function alleles) and 683 non-essential protein coding genes plus 10 small non-coding regulatory RNAs. They tested these interactions for growing *E*. *coli* in rich and minimal medium. They were able to describe the gene interaction network and the modular architecture of the *E. coli* cell envelope biosynthetic machinery.

*B*. *aphidicola* is a close relative of *E*. *coli*. In fact, most genes in *Buchnera* have their counterparts in *E*. *coli* [26]. This offers an opportunity to study the domino theory of gene loss in *Buchnera* in relation to the epistasis found in *E*. *coli*. Of course, under the assumption that epistatic interactions in *E*. *coli* are conserved, at least approximately, in *Buchnera*. In this work, we test the role played by epistatic interactions during genome reduction by gene loss among the *Buchnera*.

## 2. Materials and Methods 

*Genomic sequences*. Complete genomes from all *Buchnera aphidicola* and one *Escherichia coli* strains were retrieved from the NCBI database (ftp://www.ncbi.nlm.nih.gov/genomes/refseq/): *E. coli* str. K-12 substr. MG1655 (chromosome: NC_000913.3); *B. aphidicola* str. F009 (chr: CP002703.1, plasmid pLeu: CP002704.1); *B. aphidicola* str. G002 (chr: CP002701.1, plasmid pLeu: CP002702.1); *B. aphidicola* str. W106 (chr: CP002699.1, plasmid pLeu: CP002700.1); *B. aphidicola* str. USDA (chr: CP002697.1, plasmid pLeu: CP002698.1); *B. aphidicola* str. Ak (chr: CP002645.1, plasmid pLeu: CP002646.1, plasmid pTrp: CP002647.1); *B. aphidicola* str. JF99 (chr: CP002302.1); *B. aphidicola* str. TLW03 (chr: CP002301.1); *B. aphidicola* str. 5A (chr: CP001161.1); *B. aphidicola* str. Tuc7 (chr: CP001158.1); *B. aphidicola* str. APS (chr: BA000003.2, plasmid pLeu: AP001071.1, plasmid pTrp: AP001070.1); *B. aphidicola* str. LSR1 (chr: NZ_ACFK01000001.1); *B. aphidicola* str. JF98 (chr: CP002303.1); *B. aphidicola* str. LL01 (chr: CP002300.1); *B. aphidicola* str. Sg (chr: AE013218.1); *B. aphidicola* str. Ua (chr: CP002648.1, plasmid pLeu: CP002649.1, plasmid pTrp: CP002650.1); *B. aphidicola* str. Bp (chr: AE016826.1, plasmid pBBp1: AF492591.1); *B. aphidicola* str. BCc (chr: CP000263.1, plasmid pLeu: AY438025.1); *B. aphidicola* str. Ct (chr: CP001817.1).

*Ortholog identification*. Gene families in the 19 genomes where initially identified by using the GET_HOMOLOGUES.pl software [27]. Options used were: -M (orthoMCL algorithm), -t 0 (identify all protein families) -C 75 (minimum percent coverage in BLAST alignments) and -E 1e-05 (maximum E-value). This allowed us to identify those *E*. *coli* proteins having orthologs in *Buchnera* among the 811 protein coding genes studied by [25]. To double check identification of gene families, we performed BLAST searches of the 811 *E*. *coli* protein coding sequences against the families identified by GET_HOMOLOGUES.pl. We carefully inspected the results to check whether each one of the *E*. *coli* proteins with homologs in *Buchnera* identified by GET_HOMOLOGUES.pl is indeed the best BLAST hit. By this, we were able to identify 102 protein coding genes with clear orthologs in *Buchnera*. Of these 102 *E*. *coli* proteins, 51 were single copy orthologs and 51 had one-to-many homology relationships. In all cases, the orthology relationship identified by GET_HOMOLOGUES.pl corresponded to the best BLAST hit. In addition, there were four in-paralogs (genes undergoing duplications after speciation, in this case from the lineage leading to *Buchnera*) in *E*. *coli*: *phoE*, *ompF*, *ompC* and *ompM*. These four genes duplicated after *E*. *coli* diverged from *Buchnera* (phylogenetic analysis not shown), therefore, we considered only *phoE* for the analysis of correlated gene losses and epistasis (see below).

To inspect the conservation of *Buchnera* protein families we compared their domain composition to that of *E*. *coli* homologs. Domain composition of protein families was inferred by HMMER3 [28] and Pfam database [29]. Visualization of domain structure of protein families in a phylogentic tree was done with the ETE Toolkit [30]. The protein domain composition of each one of the protein families is shown in Appendix A.

*Phylogenetic reconstruction*. To reconstruct the phylogenetic tree of *Buchnera*, we identified all protein families sharing protein domain composition and conserved synteny. For this, we used GET_HOMOLOGUES.pl with parameters: −D − M and the option −s in the script compare_clusters.pl (the script is part of the GET_HOMOLOGUES distribution). By this, we identified 54 protein families. Next, we aligned each one of the families with MUSCLE [31] and retrotranscribed in silico the aligned protein sequences to the corresponding coding genes with PAL2NAL [32]. Phylogenetic reconstruction was done with MrBayes3.2 [33]. Parameters used were: GTR + GI model of evolution (lset nst = 6 rates = invgamma), 1e + 06 generations sampling every 500 (mcmc ngen = 1,000,000 samplefreq = 500 printfreq = 500 diagnfreq = 5000) and discarded 25% of samples from the cold chain (relburnin = yes burninfrac = 0.25). The analysis converged (average standard deviation < 0.01). To summarize the tree, we discarded 25% of the samples: sumt relburnin = yes burninfrac = 0.25. Eleven nodes had a posterior probability of 1.0 and four nodes had a posterior probability > 0.9. The phylogenetic tree with posterior probabilities is shown in Appendix A.

*Analysis of correlated gene loss*. To identify pair of genes that were lost in a correlated fashion along *Buchnera* evolution, we used the algorithm *Discrete* from the software BayesTraitsV2 [34,35]; http://www.evolution.rdg.ac.uk/BayesTraitsV2.html). For that purpose, we used the phylogenetic tree of *Buchneras* and *E*. *coli* reconstructed with MrBayes3.2 and a matrix of the presence/absence of genes per gene family per genome. This matrix was reconstructed from the results of GET_HOMOLOGUES.pl and consisted of 85 gene families (rows) per the 19 genomes (columns). Briefly, “Discrete” algorithm tests for correlated character evolution (codified by two binary traits) along a phylogenetic tree. It does so by comparing two models. In the first model, characters are assumed to evolve independently. In the second model, it is assumed that the characters evolve in a correlated fashion. In our case, the characters were presence (1) or absence (0) of a given gene in a gene family. We compared whether the pattern of gene losses in a given gene family indicates correlated evolution with the pattern of gene loss in another family. The models are compared by a likelihood ratio test (LRT) which is asymptotically distributed as a χ^2^ distribution with the number of degrees of freedom equal to the difference of the number of parameters between the two models (df = 8 − 4). Since we incurred in multiple testing (we tested for correlated evolution between 85 gene families) we applied used false discovery rate (FDR) to control for false positives. In the first case, we applied an FDR of 10% to the list of *p*-values, and in the second, case we used an FDR of 45%. In our analysis, an FDR of 45% correspond approximately to *p*-values < 0.05. 

*Epistasis analysis*. In order to study the relationship between epistasis and correlated gene loss, we compared our results of correlated gene pairs to those reported for *E*. *coli* by [25]. We assume that *E*. *coli* represents an “ancestral state” for functional gene interactions. This is based on the fact that the gene content of *Buchnera* is almost a subset of *E*. *coli* the gene content [26]. Specifically, empirical gene interactions data of *E*. *coli* was obtained from the Appendix A of [25]. We used the same criteria to define positive and negative interactions as [25]. Positive and negative interactions are defined according to an E-score threshold. All empirical interactions where catalogued as positive epistasis if the E-score was above 2 and as negative epistasis if the E-score was below −2. Scores between −2 and 2 where considered as neutral. In Appendix A from [25], there are 235,031 interactions in rich medium and 129,313 interactions in minimal medium. Due to the experimental procedure, the interactions in minimal medium are a subset of the interactions in rich medium. We compared our results of correlated gene loss analysis to all 129,313 interactions from minimal medium and to the corresponding 129,313 interactions from rich medium.

The gene interaction web was implemented with Cytoscape3.5.1 [36]. The synteny map was done with genoPlotR [37]. All statistical analyses were done in R (https://www.r-project.org).

## 3. Results

### 3.1. Ortholog Identification between Buchnera and E. coli

Among the genes studied by [25], we identified 102 with clear orthologs in *Buchnera* (Appendix A). These genes participate in diverse pathways in *E*. *coli* (Figure 1). Of these 102 families, 17 are conserved between *E*. *coli* and all 18 *Buchnera* genomes. The remaining 85 families show varying degrees of conservation among *Buchnera* providing information for testing the role of epistasis in relation to the domino theory of gene loss.

The numbers of gene losses vary between families and genomes. In some of these families, there is only a single gene loss, while in other families there are as many as 17 gene losses (Figure 2). Most of the families show 4 or fewer gene losses. In addition, gene losses among these 85 gene families occur unevenly among *Buchnera* genomes. As shown in Figure 3, *B*. *aphidicola* Cc and *B*. *aphidicola* Ct accumulate most gene losses. These are followed by *B*. *aphidicola* JF98 and *B*. *aphidicola* Bp.

### 3.2. Essential Genes in E. coli Tend to be Retained in the Genomes of Buchnera 

As expected, we found that genes that are reported as essential in *E*. *coli* are less likely to belong to families showing gene losses among the *Buchnera*. The odds ratio of this association is 2.7 and the *p*-value is 0.06 (95% CI = 0.95 to ∞, one sided Fisher exact test). Although the probability of finding this association at random is slightly larger than 0.05, this shows a trend that suggests that essential genes in *E*. *coli* tend to be essential also in *Buchnera*, despite the differences in biology and environment between the two species. It also suggests that gene function in *E*. *coli* is conserved among *Buchnera*, i.e., lessons learned in *E*. *coli* can be extrapolated to *Buchnera* with some degree of confidence.

### 3.3. Correlated Gene Losses along Buchnera Phylogeny

To investigate the relationship between correlated gene loss and epistasis, we first studied if some of the 85 variable families show evidence of coevolution. Using the BayesTraits software, we estimated the probability that the pattern of gene conservation between two given families is the result of random gene losses. If this probability is smaller than 0.05, we assumed that the similarity in gene losses is unlikely to be due to chance alone. That is, genes were lost in a correlated fashion, presumably because their protein products interact [35]. 

BayesTraits analysis identified 356 pairs of gene families whose similarity in their pattern of gene losses along the phylogeny of *Buchnera* would be found by chance less than 5% of the time (*p*-value < 0.05). Of course, this probability is under the null hypothesis of no correlated gene losses. These 356 pairs are conformed by 57 gene families. 

Since we performed several statistical tests (i.e., we calculated the probability of correlated evolution, under the null hypothesis of no association, between all the pairs that can be formed with 85 gene families), we expect more false positives than those specified by a *p*-value of 0.05. Therefore, it is necessary to introduce a correction to the *p*-value. In this case, we used the false discovery rate (FDR or Benjamini-Hochberg procedure). When controlling for a 10% (FDR < 0.1) of false positives, the above numbers reduced our original results to 43 pairs of correlated gene losses conformed by 13 gene families. These gene families and their correlations are shown in Figure 4. It is important to notice that none of these genes are found adjacent to each other in *Buchnera* genomes (Figure 5), indicating that correlated gene loss is not simply due to proximity within the genome. We suggest that most pairs of gene families showing correlated gene loss (as defined by a *p*-value < 0.05 and FDR < 0.1) code for proteins that functionally interact.

### 3.4. Is Correlated Gene Loss in Buchnera Explained by Epistasis Data in E. coli?

We then decided to investigate whether the correlations identified with BayesTraitsV2 in *Buchnera* are associated with known epistatic interactions among these genes in *E*. *coli* [25]. In particular, we would expect positive (or alleviating) epistasis to facilitate correlated gene losses (due to reduction in selective costs), while negative (or aggravating) epistasis would impair correlated gene losses (because of increased selective costs). If so, correlated gene losses in *Buchnera* should occur more often among gene families whose *E*. *coli* orthologs show positive epistasis and if correlated gene losses among *Buchnera* occurs less often among gene families whose *E*. *coli* orthologs show negative epistasis (Figure 6a). 

We tested the above-mentioned hypotheses by using epistasis data from [25]. The data, was obtained from growing *E*. *coli* in minimal and rich medium. We found that for minimal medium, the probability of observing actual data (Figure 6b) is less than 5% (*p*-value ≈ 0.02) under the null hypothesis of no association between correlated gene lose and epistasis (Table 1). We find the same result if we relax our false discovery rate to FDR < 0.45, this to include all *p*-values from BayesTratisV2 analysis smaller than 0.05. That is, we found that the fraction of gene pairs showing positive epistasis *versus* negative epistasis is different in gene pairs showing correlated gene loss than in those not showing it. In other words, positive epistasis seems to favor correlated gene losses while negative epistasis seems to disfavor it (see below). However, it is important to notice that we did not find this association for epistasis data obtained from *E*. *coli* growing in rich medium. 

It is intriguing that we only find the association between epistasis and gene loss for minimal and not for rich medium data. In *E*. *coli*, essential genes showed more positive epistatic interactions than non-essential genes in rich medium. While in minimal medium, essential genes showed more negative epistatic interactions than non-essential genes [25]. Our analysis mirrors the pattern (Figure 7 and Appendix A). Essential genes also showed more positive interactions in rich medium (148/34 ≈ 4.4) than non-essential genes (60/52 ≈ 1.2); and in minimal medium, essential genes showed more negative interactions (66/196 ≈ 0.3) than non-essential genes (69/58 ≈ 1.2). Interestingly, non-essential genes showed about the same proportion of positive versus negative epistasis in rich and minimal medium. This suggests that the *p*-value differences between rich and minimal medium, reported in Table 1, are driven mostly by changes in the proportion of positive versus negative epistatic interactions in essential genes. According to [25], the increase of negative epistatic interactions between essential genes in minimal medium suggest the emergence of compensatory relationships under environmental constrains. That is, under minimal medium there are less redundant pathways. This decrease of redundant pathways causes a concomitant increase of negative interactions.

## 4. Discussion

### 4.1. Conservation of Epistatic Relationships between E. coli and Buchnera

The extrapolation of the experimental results found by [25] to *B*. *aphidicola* depends on several assumptions. First, orthologs have to be accurately identified, and the functions of these genes in *B*. *aphidicola* have to be the same as in *E*. *coli*. As mentioned before, ortholog identification between *E*. *coli* and *B*. *aphidicola* is relatively straightforward because of the evolutionary distance between these species and the high level of synteny among *Buchnera*. However, it is not always easy to evaluate if the ortholog gene in *B*. *aphidicola* is performing the same function as in *E*. *coli* or if it is functional at all. Often, the ortholog in *B*. *aphidicola* is lacking some domains present in the *E*. *coli* version of the protein. For instance, of all the 85 families studied here, only in 70 of them the orthologs in *B*. *aphidicola* have exactly the same domain composition in *Buchnera* as the *E*. *coli* proteins (Appendix A). Despite this lack of conservation of protein domains, essential genes in *E*. *coli* tend to be more conserved in *B*. *aphidicola* than non-essential genes (*p*-value = 0.06). This suggests that protein function tend to be conserved. The *p*-value of this association is slightly larger than 0.05. However, there is ample evidence showing that essential genes tend to be conserved in evolution more than non-essential genes [38,39,40,41,42,43]). Therefore, we consider that a *p*-value of 0.06 is in agreement with previous results. 

This leads us to the second point, whether epistatic interactions in *E*. *coli* can be extrapolated to *B*. *aphidicola*. This is more difficult to evaluate, but existent data suggest that epistatic interactions are conserved to some degree between species sharing similar biology. For instance, *Schizosaccharomyces pombe* and *Saccharomyces cerevisiae* share between ~18% to 29% of their negative (synthetic lethal) genetic interactions [44,45]. This percentage increases above 50% for positive interactions when the involved genes code for physically associated proteins [44,46]). Remarkably, these interactions are shared despite several million years of divergence. The time of divergence between *S*. *pombe* and *S*. *cerevisiae* has been estimated to be from ~400 to ~1000 million years [47,48]. In comparison, the divergence time between *B*. *aphidicola* and *E*. *coli* is estimate to be between ~160 to 280 million years [49,50]). Therefore, it is likely that similar percentages of shared epistatic interactions are shown between the bacteria analyzed here.

Given all the difficulties discussed above associated to extrapolating epistasis data from *E*. *coli* to *B*. *aphidicola*, we consider that there is statistical evidence (Table 1) indicating that correlated gene loss occurs more often among gene pairs showing positive epistasis than among those showing negative epistasis. This indicates that the domino effect applies mostly to genes participating in the same pathway or protein complex [51]. Genes in the same pathway or protein complex tend to show positive epistasis.

It is important to notice that the domino theory, to better work as a hypothesis, requires a refinement. Ultimately, the selective advantage conferred by a given gene is dependent on the environment in which the bacteria live. Therefore, a gene loss mutation cannot by itself cause the relaxation in selective pressure of functionally related genes. To reach fixation, a domino-effect-gene-loss requires population bottlenecks to get rid of bacteria lacking the mutation by chance. Otherwise, bacteria that have retained the gene might outcompete the mutant strain. Alternatively, the domino-effect-gene-loss has to provide some selective advantage. In intracellular mutualistic bacteria, gene-loss mutations favoring the fitness of the holobiont will be favored by selection.

### 4.2. The Role of Epistasis in Evolution of Genome Reduction

“One of the more interesting long term questions in evolutionary biology is whether or not epistasis determines the path of evolutionary change” [24]. Here, we provide statistical support for the role played by epistasis in correlated gene loss in *B. aphidicola*. But, why is this interaction only is found only for epistasis data obtained from minimal and not from rich medium? If we consider that: (a) the relationship between epistasis and correlated gene loss is unlikely due to chance (*p*-value = 0.02, FDR = 10%) and (b) accept that negative epistasis impairs and positive epistasis facilitates gene loss. This lead us to conclude (c) that somehow the medium on which *B. aphidicola* thrives causes a pattern of epistatic relationships similar to those of *E. coli* growing on minimal medium.

This result may seem counterintuitive. One would expect that intracellular conditions would be more similar to rich medium. However, *B*. *aphidicola* does not live in direct contact with the cytoplasm of the host. On the contrary, *B*. *aphidicola* grows inside symbiotic vesicles within bacteriocytes whose membranes are of host origin [3,52]. Therefore, the host seems to have direct control of what molecules are provided to *B*. *aphidicola* for nutrition. In fact, recent studies show that *B*. *aphidicola* population is carefully controlled along aphid life cycle [53] and that *Buchnera* may grow under nitrogen limiting conditions [9].

A more appealing explanation of this pattern is that in the reduced proteome of *B*. *aphidicola* there are few, if any, redundant pathways. Therefore, aggravating interactions will be more common than alleviating ones and, as mentioned above, this cellular condition is similar to that of *E*. *coli* growing on minimal medium, where essential genes showed more negative epistatic interactions than non-essential genes [25].

How widespread are the effects detected here on bacterial genome evolution? In a recent study [54] calculated the frequency of pairs of ortholog genes that co-occur across ~600 species of bacterial genomes. They also identified those pairs of orthologs that tend not to occur in the same genome. They called the former “correlogs” and the later “anti-correlogs”. They found that correlogs in these genomes organize in 483 subgroups. These subgroups associate to each other by “anti-correlogy”. Because genes within subgroups tend to participate in the same functional category it is likely that epistasis among them will tend to be positive, and that they will be retained by natural selection or, according to the results shown here, lost together once their function becomes dispensable.

## 5. Conclusions

*B*. *aphidicola* allow us to study the relationship between epistasis and the evolutionary fate of genes–under the assumption that epistatic interactions in *E*. *coli* are maintained in *B*. *aphidicola*. We provided statistical evidence indicating that, along genome reduction in *B*. *aphidicola*, positive or alleviating epistasis facilitates correlated gene loss, while negative or aggravating epistasis impairs correlated gene loss. Our analysis paves the way to more extensive analysis of the role of epistasis in genome evolution in *B*. *aphidicola* by including more genes [55,56], other factors like protein-protein interactions [57], metabolic networks [58], or by studying this phenomenon in other model organisms like yeast.

## Figures and Tables

**Figure 1 life-08-00017-f001:**
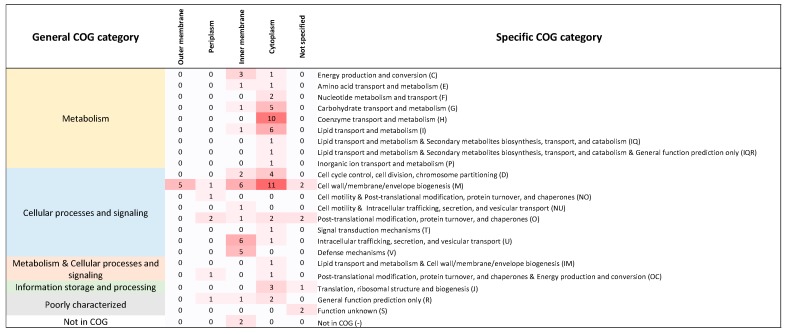
Functional classification of 102 genes from *E*. *coli* with orthologs in *Buchnera*. Each cell denotes the number of genes classified in each COG category. We also show the localization of the corresponding protein product in four cellular compartments. The classification of these genes follows that of Babu et al. (2011).

**Figure 2 life-08-00017-f002:**
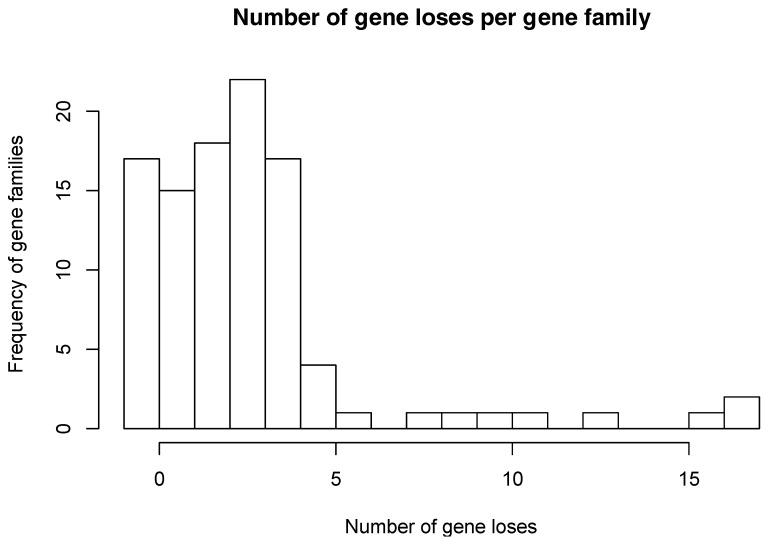
Distribution of number of gene losses per gene family. For each one of the 102 gene families studied here, we counted the number of gene losses among the 18 *Buchnera* genomes.

**Figure 3 life-08-00017-f003:**
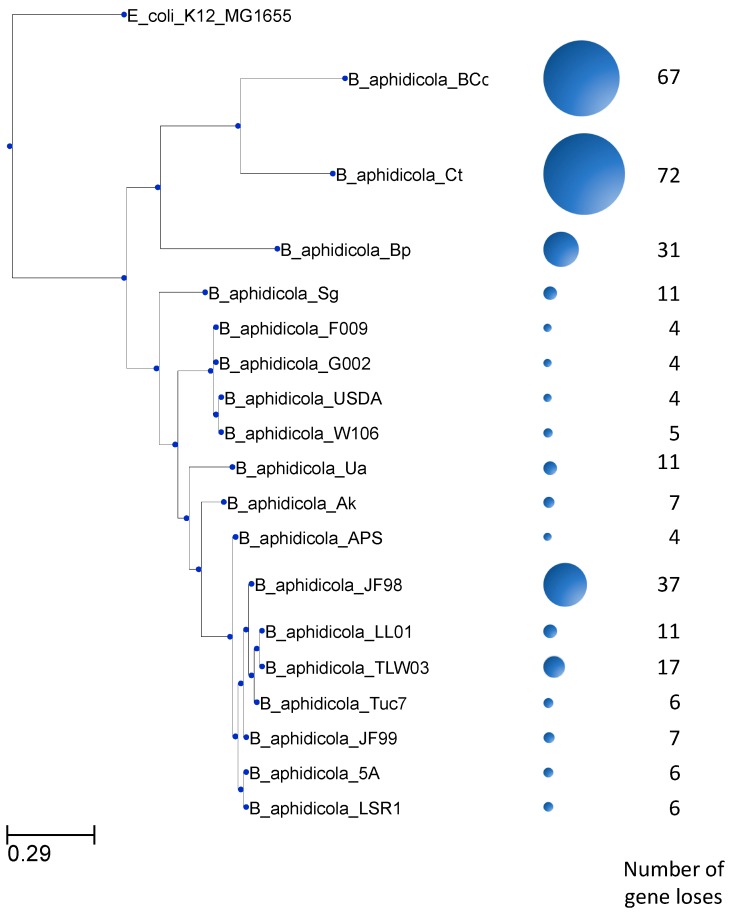
Distribution of gene losses among *Buchnera* genomes. The number of gene losses in the 102 gene families studied here is unevenly distributed among *Buchnera* genomes. The size of the circles corresponds to the number of gene losses.

**Figure 4 life-08-00017-f004:**
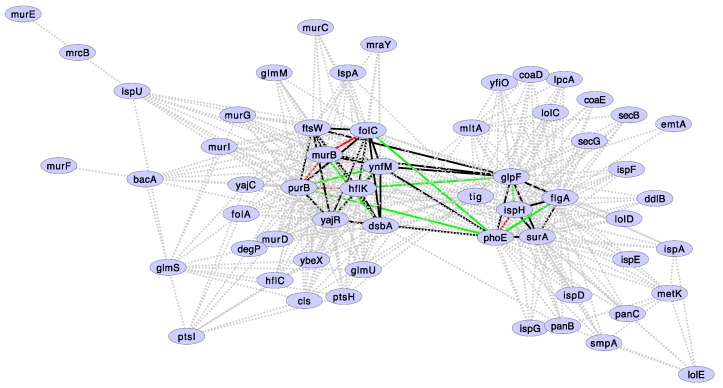
Gene families showing correlated gene losses along *Buchnera* phylogeny. Doted lines: Correlated gene losses showing a *p*-value < 0.05. Solid lines: Correlated gene losses discovered by setting a FDR < 0.1. Green lines: Positive epistatic interactions in minimal medium. Red lines: Negative epistatic interactions in minimal medium. Epistatic interactions are according to Babu et al. (2011). For clarity, epistatic interactions are shown only for those correlated gene losses showing an FDR < 0.1. An FDR < 0.1 captures all interactions with a *p*-value < 0.0015.

**Figure 5 life-08-00017-f005:**
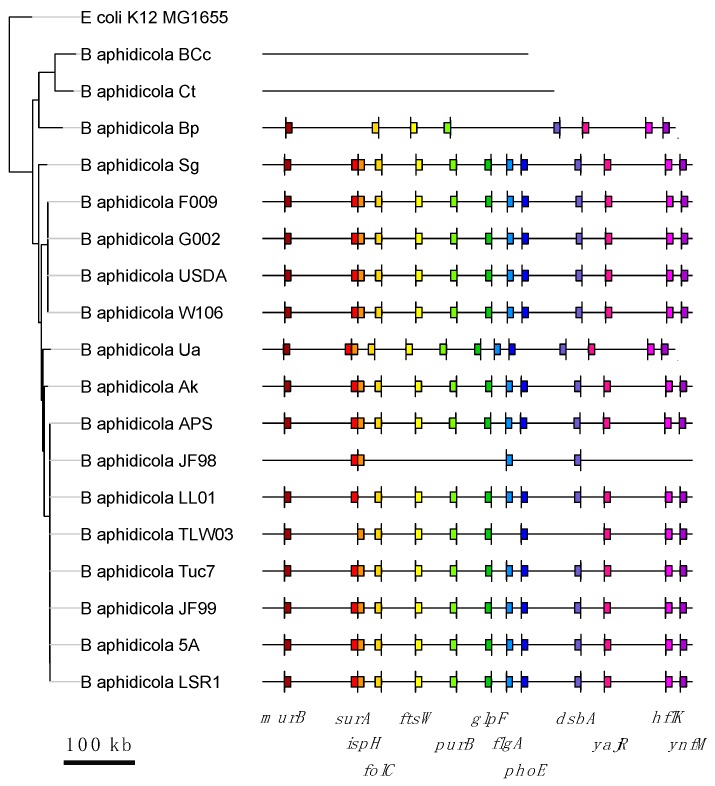
Localization of genes showing correlated gene loss (FDR < 0.1) along *Buchnera* genomes. The size of the genomes is shown with horizontal lines and the relative localization of the genes showing correlated gene lose with colored squares. The vertical lines in each one of the genes indicate the direction of transcription. For clarity, the genome of *E*. *coli* is not shown. None of the genes are coded contiguously. *surA* and *ispH* look contiguous because of the scale, but are separated by more than 7000 bp.

**Figure 6 life-08-00017-f006:**
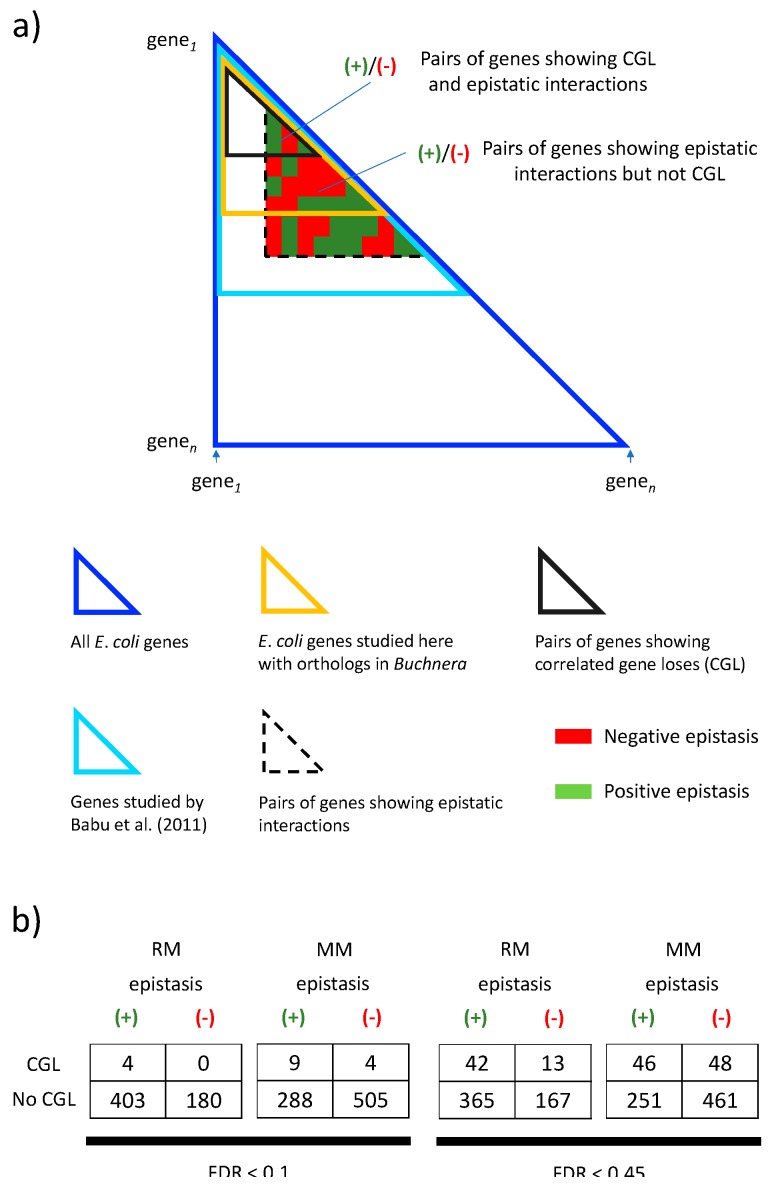
Comparison of correlated gene loses (CGL) identified in *Buchnera* with epistatic interactions in *E*. *coli* detected by Babu et al. (2011). (**a**) Dark blue triangle: All *E*. *coli* gen pairs are accommodated in an n x n matrix (only “half” the matrix is needed, below its main diagonal). Light blue triangle: Pairs of genes studied by Babu et al. (2011). Gold triangle: Pairs of *E*. *coli* genes with orthologs in *Buchnera* studied here. Gray triangle: Pairs of genes showing correlated gene lose in *Buchnera*. Dashed triangle: Pairs of genes showing epistatic interactions by Babu et al. (2011). Among the set of genes showing epistatic interactions in *E*. *coli*, some have orthologs in *Buchnera* showing CGL and some of them do not. (**b**) Frequency of gene families showing positive and negative epistatic interactions and showing (or not) CGL. The frequency is shown for an FDR < 0.45 and for a FDR < 0.1. The frequency is also shown for minimal and rich media. An FDR < 0.45 comprises all CGL showing a *p*-value < 0.05. The size of the triangles is not to scale.

**Figure 7 life-08-00017-f007:**
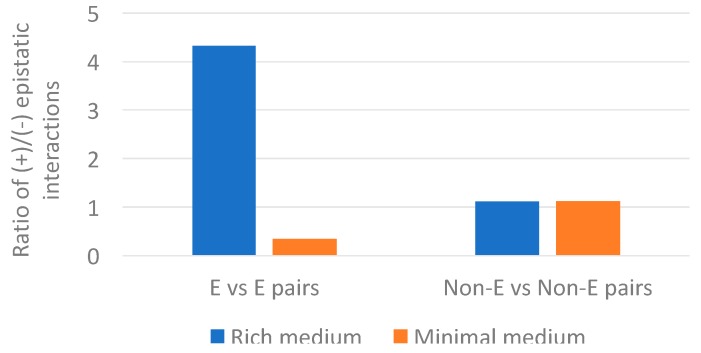
Ratio of positive versus negative epistatic interactions for pairs of essential (E) genes and pairs of non-essential (Non-E) genes in rich and minimal medium. The epistasis data is from *E*. *coli* genes having clear orthologs in *Buchnera* genomes.

**Table 1 life-08-00017-t001:** The proportion of gene pairs showing positive and negative epistasis is different among the set of gene pairs showing correlated gene loss and those not showing it. However, the probability of finding this association under the null hypothesis of no association is less than 0.05 only for epistasis data from growing *E*. *coli* in minimal medium. The probability was calculated with a two-sided Fisher exact test. We calculated the probability for two FDR cutoffs: an FDR < 0.1 comprises all pairs of genes showing a *p*-value < 0.01 under the null hypothesis of no correlated gene lose and an FDR < 0.45 comprises all pairs of genes with a *p*-value < 0.05 under the null hypothesis of no correlated gene lose. In all cases, the null hypothesis is True odds ratio is greater than 1. RM, rich medium; MM, minimal medium.

Correlated Losses Gene Set Defined by	Media	*p*-Value	95% CI	Odds Ratio
FDR < 0.1	RM	0.31	0.29 to ∞	Inf.
(*p*-value < 0.01)	MM	0.02	1.09 to 17.7	3.93
FDR < 0.45	RM	0.28	0.75 to 3.08	1.48
(*p*-value < 0.05)	MM	0.01	1.11 to 2.78	1.76

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
