# Peer review of "Testing the Domino Theory of Gene Loss in Buchnera aphidicola: The Relevance of Epistatic Interactions"

_life, 2018, doi:10.3390/life8020017_

Round 1

Reviewer 1 Report

In this paper, gene losses in  species of Buchnera were studied in relation to epistasis in E. coli (where positive epistasis means that the combined beneficial effect of two mutations is greater than the sum of their effects if these were independent). The authors find that when a particular gene/product is lost,a second gene/product that interacts with the first is more likely to be lost if its loss alleviates the loss of the first gene/product (corresponding to positive epistasis).The results are interpreted in terms of the authors’ “domino theory of gene loss” which attributes genome reduction in Buchnera to the paucity of redundant metabolic pathways.

The work described appears solid. The authors examined 102 protein-coding genes in E. coli with orthologs in Buchnera. The conservation of Buchnera protein families was obtained by comparing their domain composition with that of E. coli homologs and the phylogenetic tree of Buchnera was constructed by identifying all protein families sharing protein domain composition and proximity/order on the chromosome. Pairs of genes lost together during the evolution of Buchnera were discovered using an alorithm that compares the evolution of characters when they are independent with when they are correlated. To study the relationship between epistasis and correlated gene loss,they compared their results of correlated gene loss analysis to over 129,000 interactions from rich and minimal media.

This is an interesting and important paper that is, in principle, suitable for Life. It could, however, be made more accessible for readers who come from disciplines other than evolutionary microbiology. A number of largely minor comments follow.

68 ‘extreme synteny’ say briefly what this is. Note that, in the solenoid model for chromosome structure (e.g., Bouyioukos et al. Sci Rep 2016; 6: 27978), the function of the linear proximity of one gene to another can be preserved in 3D by the positioning of the genes to one another at multiples of a particular period.

68 ‘and almost total lack of horizontal gene transfer’. In this context, they might stress (as the authors know) that Buchnera do have plasmids and it is conceivable that these play a role in the gene loss that interests them.

75 ‘versus the expected fitness effects of mutations’ – perhaps add ‘where these mutations are independent’. In talking about the Domino theory and epistasis, they might like to refer to extragenic suppressors. Mutants in which DnaA, for example, is inactivate can survive if there is a compensatory mutation in RnaseH. The point is that a mutation in one gene leading to a mutation in another can be more than a passive process allowing genetic drift: it can actively drive selection.

82-87 ‘Babu ... tested these interactions ...’ So what? There seems to be a conclusion missing.

121 ‘there were four in-paralogs in E. coli: phoE, ompF,ompC and ompM.’ Perhaps add for the non-specialist: ‘(i.e., these four genes are copies of one another)’.

163 ‘We assume that E. coli represents an“ancestral state” for functional gene interactions.’ What is the justification for this assumption – is it always justified?

 238 ‘In other words, as expected, positive epistasis seems to favor correlated gene losses while negative epistasis seems to disfavor it. However, it is important to notice that we did not find this association for epistasis data obtained from E. coli growing in rich medium.’ This is a bit cryptic. Since they discuss it later, add ‘(see below)’

 312 ‘Otherwise,bacteria lacking the gene loss might outcompete the mutant strain. Try ‘Otherwise, bacteria that have retained the gene might outcompete the mutant strain’

 313 ‘Alternatively, the domino-effect gene-loss has to provide some selective advantage’.A general explanation for such an advantage would be that it avoids the unnecessary expression of genes, which constitutes a burden. Another explanation would be that many enzymes and other proteins form assemblies and filaments in which the absence of one of the constituents might lead to an incomplete structure that would be defective in signalling and perhaps toxic.In interpreting their results, the authors might therefore consider the relevance of the number of types of enzymes participating in an assembly. They might also consider ‘moon-lighting’.

The figures lack legends.

Figure 1 shows six bars each containing ‘cell division’ in its label. One of these bars is simply labelled ‘cell division’ – does it contain different genes from those in the other bars? Say what ECA and LPS stand for.

Figure 2 what are these families? ‘number of gene losses’ (correct this in Figure 3 too)

Figure 5 what do the symbols mean? Do the lines represent chromosomes?

Author Response

Thanks for the review. 

68 ‘extreme synteny’ say briefly what this is. Note that, in the solenoid model for 

chromosome structure (e.g., Bouyioukos et al. Sci Rep 2016; 6: 27978), the function of the 

linear proximity of one gene to another can be preserved in 3D by the positioning of the 

genes to one another at multiples of a particular period.

Response

We clarified what we mean by 'extreme synteny'.

68 ‘and almost total lack of horizontal gene transfer’. In this context, they might stress 

(as the authors know) that Buchnera do have plasmids and it is conceivable that these play 

a role in the gene loss that interests them.

Response

We included plasmids in the analysis. However, all cases of significative gene losses were

from genes coded in the chromosome.

75 ‘versus the expected fitness effects of mutations’ – perhaps add ‘where these mutations 

are independent’. In talking about the Domino theory and epistasis, they might like to 

refer to extragenic suppressors. Mutants in which DnaA, for example, is inactivate can 

survive if there is a compensatory mutation in RnaseH. The point is that a mutation in one 

gene leading to a mutation in another can be more than a passive process allowing genetic 

drift: it can actively drive selection.

Response

It is an interesting observation. We added 'where these mutations are independent'.

82-87 ‘Babu ... tested these interactions ...’ So what? There seems to be a conclusion 

missing.

Response

We added a sentence describing the main finding of Babu et al. (2011).

121 ‘there were four in-paralogs in E. coli: phoE, ompF,ompC and ompM.’ Perhaps add for 

the non-specialist: ‘(i.e., these four genes are copies of one another)’.

Response

We added a sentence to clarify the concept of 'in-paralogs'.

163 ‘We assume that E. coli represents an“ancestral state” for functional gene 

interactions.’ What is the justification for this assumption – is it always justified?

Response

We added a sentence to clarify our assumption.

238 ‘In other words, as expected, positive epistasis seems to favor correlated gene 

 losses while negative epistasis seems to disfavor it. However, it is important to notice 

 that we did not find this association for epistasis data obtained from E. coli growing in 

 rich medium.’ This is a bit cryptic. Since they discuss it later, add ‘(see below)’

Response

We added (see below) and deleted 'as expected'. Thanks for the suggestion.

 312 ‘Otherwise,bacteria lacking the gene loss might outcompete the mutant strain. Try 

 ‘Otherwise, bacteria that have retained the gene might outcompete the mutant strain’

Response

We have made the change. Thanks for the suggestion.

 313 ‘Alternatively, the domino-effect gene-loss has to provide some selective advantage’.

 A general explanation for such an advantage would be that it avoids the unnecessary 

 expression of genes, which constitutes a burden. Another explanation would be that many 

 enzymes and other proteins form assemblies and filaments in which the absence of one of 

 the constituents might lead to an incomplete structure that would be defective in 

 signalling and perhaps toxic. In interpreting their results, the authors might therefore 

 consider the relevance of the number of types of enzymes participating in an assembly. 

 They might also consider ‘moon-lighting’.

Response

We added a sentence to the end of the paragraph.

The figures lack legends.

Figure 1 shows six bars each containing ‘cell division’ in its label. One of these bars is 

simply labelled ‘cell division’ – does it contain different genes from those in the other 

bars? Say what ECA and LPS stand for.

Figure 2 what are these families? ‘number of gene losses’ (correct this in Figure 3 too)

Figure 5 what do the symbols mean? Do the lines represent chromosomes?

Response

We added figure legends. We apologize for this. We also updated Figure 1 - now is more

informative. We also added two new supplementary tables (Table S2 and S3). These tables 

contains pairs of genes showing epistasis and correlated gene losses. 

Reviewer 2 Report

Martinez Cano et al. present an interesting study on genome evolution (i.e. gene-loss due to endosymbiontic life style) using genetic interaction data to support the 'domino' theory of genome reduction. The study is based on a comparison between E. coli and Buchnera. For these, orthologous relationships can be clearly established, which is crucial to the study. Also, these bacteria are closely enough related to assume conservation of genetic interactions.

The assumption tested is that alleviating interactions (which tend to occur within complexes/pathways) should be preferably be lost, as predicted by the domino theory. Indeed, this trend is clearly visible, especially in a genetic interaction data-set from minimal medium. (Indicating that this condition reflects the special endosymbiontic life style of Buchnera better).

The study is well done, scientifically and statistically sound. I think it is a great application of the genetic interaction data on questions of evolution.

- I'm missing figure legends?! Please improve on that/add them

- I would add a supplementary table with the actual interactions and their evolutionary fate. (GIs following the domino behavior might be interesting interactions to follow up on, since they are more likely to be true positive GI measurements)

- what do the numbers like '499198529' in the supplementary table mean?

- perhaps some argumentation why not more genetic interaction data has been used (good argument would be that minimal media is needed + that the Babu et al. 2014 study is based on randomly selected genes not covering sufficient complete cellular processes

optional:

- test of domino theory with physical interactions? Which of the GI could be explained by PPI?

-Are there genetic interactions (how many) and subsequent 'domino events' that could not be seen by using pathway or PPI data?

- probably there is not enough data for this, but it would be interesting if there is a trend for common loss with increasing strength of the alleviating interaction

Author Response

Thanks for the review. 

We added figure legends. We apologize for this. We also updated Figure 1 - now is more informative. We also added two new supplementary tables (Table S2 and S3). These tables contains pairs of genes showing epistasis and correlated gene losses. 

Numbers like '499198529' correspond to NCBI GI protein numbers. We added a legend to the end of the Table S1.

We would like very much to extend our analysis to more epistasis as well as PPI data. We will do it in the near future.